# Service network design for freight railway transportation: The Chinese case

**Xiaoyu Xin** [ID] *

School of Traffic and Transportation, Beijing Jiaotong University, Beijing, China

* 19114058@bjtu.edu.cn

## Abstract

Service network design is a typical tactical planning problem faced in freight railway transportation. In this paper, we propose an integer linear programming model for the rail freight service network design problem in the context of the China railway system. The model aims to minimize the overall costs (including train costs and car costs) while satisfying a set of practical restrictions and operational requirements. Our model also considers a distinct operational feature in Chinese railway transportation practice, i.e., the tree-shaped path. A numerical experiment as well as a real-life case study from the China Railway Jinan Group (CR Jinan) is conducted to demonstrate the proposed solution approach. The real-life problem size reaches as many as 139 stations, 473 alternative train services and 562 shipments. Computational results show that our proposed approach is able to obtain quality solutions within a reasonable time frame.

**Data Availability Statement:** All relevant data are within the manuscript.

**Funding:** The author(s) received no specific funding for this work.

## Introduction

Rail freight transportation is an important logistics system that handles the movement of millions of railcars to support the global and regional economy. In 2020, despite the outbreak of the COVID-19 epidemic, the China railway system carried 4.55 billion tons of freight, generating 3,051.45 billion cargo ton-kilometers. The increasing freight demand, operational costs (such as labor costs) and network complexity have boosted the development of operations research (OR) methods applied to railway transportation [1–3]. One of the most relevant optimization problems in railway transportation is the service network design problem, that is, the problem of designing the set of origin-destination train services between yards of the network. This is a typical tactical planning problem, as the effects of decisions span a mid-term planning horizon (approximately 6–12 months) [4].

Generally, if a shipment's volume is not large enough to organize a direct train, the shipment is transported to its final destination by a sequence of train services rather than by a single direct train service. This transportation practice involves two important decision-making processes: i) between which pair of stations should direct train services be provided and with what frequencies; and ii) to which train service(s) should a shipment be assigned from the shipment's origin to the destination. The former process is usually considered when designing the railroad blocking plan (also called the train makeup plan). The blocking plan aims to determine the overall blocks to be built at each yard and the specific shipment that should be placed

**Competing interests:** The authors have declared that no competing interests exist.

into each block to reduce the intermediate handlings as they travel from their origins to their respective destinations [5–11]. The latter process, i.e., to which train service(s) should a shipment be assigned from the shipment's origin to destination, is defined as the block-to-train assignment problem in the scientific community [6–15]. Since the railroad blocking problem and block-to-train assignment are highly interrelated, some researchers consider several of the issues as an integrated service network design problem. Crainic [16] and Yaghini and Akhavan [17] presented a state-of-the-art review of service network design modeling efforts and mathematical programming developments for network design. A new classification of service network design problems and formulations was also introduced. Crainic and Rousseau [18] presented a general modeling framework based on a network optimization model, which may be used to assist and enhance the tactical and strategic planning process for such a system. The problem was solved based on decomposition and column generation principles. Andersen et al. [19] presented a new optimization model for the tactical design of scheduled service networks for transportation systems where several entities provide critical service and internal exchanges and coordination with neighboring systems. Lulli et al. [4] presented a case study on freight railway transportation in Italy, which was a byproduct of research collaboration with a major Italian railway company. They highlighted the main features of the Italian reality and proposed a customized mathematical model to design the service network. Zhu et al. [20] addressed the service network design problem for freight rail transportation. A comprehensive model integrating service selection and scheduling, car classification and blocking, train makeup, and routing of time-dependent customer shipments based on a cyclic three-layer space-time network representation of the associated operations and decisions and their relations and time dimensions was proposed. Duan et al. [21] presented a new frequency-based service network design model with transshipments, capacity constraints and heterogeneous users. They also applied the model to demonstrate that including heterogeneity explicitly in network design pays off in terms of an improved user performance of the network. The service network work design problem is also an attractive topic in the fields of express shipment delivery [22–24], liner shipping [25–27] and air cargo transportation [28–30].

A large portion of the papers listed above refer to the North American and European freight railway transportation systems. Although the decision problems are similar in nature, there are several differences among Chinese, North American and European freight railway transportation companies, both in the way of managing operations and in the way of handling tactical planning. For example, railway operations are supervised by the China Railway with a hierarchical management system, which is different from the parallel management system in North America. The freight train operation organization strategy of the China Railway is quantity oriented and the dispatch of train services depends heavily on freight volume. While in the European railway system, the freight train operation organization strategy is quality oriented and the dispatch of train services is based exactly on train timetable. These two basic types of freight train operation organization strategies are respectively defined as the organized type and the planned type by [31]. Moreover, the Chinese railway system has a distinct character with respect to train routing, i.e., the tree-shaped path. Unlike other transportation modes, the physical path of traffic flow in a rail system has its own unique characteristics, among which the most particular is that shipments gradually converge into one single traffic flow and then travel together through a tree-shaped path. More specifically, when two or more shipments arrive at a railyard, those intended for the same destination are treated as one single freight flow and are shipped through the same path during the remainder of the trip, regardless of their origins. Consequently, mathematical models developed for the European and North American systems may not be appropriate for the Chinese scenario. Therefore, this paper aims to make the following contributions to the literature.

1. The rail freight transportation in the China railway system is analyzed in detail. We describe the rail freight flow organization process for both express goods and nonexpress goods in the context of China railway practice. On this basis, we introduce the freight train service network design problem and summarize the main features of the problem. Moreover, the tree-shaped path, which is a distinct operation feature in Chinese rail freight transportation, is also proposed.

2. An integer linear programming model for the rail freight service network design problem in the context of the China railway system is developed. The model aims to minimize the overall costs (including train costs and car costs) while satisfying a set of practical restrictions and operational requirements, including nonsplitable flow constraints and transportation time limit constraints. A distinctive constraint considered in the mathematical model is the tree-shaped path, which commonly appears in the context of China railway freight transportation. Moreover, the model takes both express goods and nonexpress goods into account.

3. A real-world case study from China Railway Jinan Group Co., Ltd. is carried out to examine the validity of the method. The real-life problem size reaches as many as 139 stations, 473 alternative train services and 562 shipments. Comparisons with manual solutions indicate that our solution approach outperforms the manual method in terms of both solution quality and computational speed. The computational results can be used directly by railway planners to save painstaking efforts in making such plans.

The remainder of this paper is organized as follows. Section 2 gives a detailed description of Chinese freight railway transportation. An integer linear programming model for the rail freight service network design problem in the context of the China railway system is proposed in Section 3. Section 4 conducts a numerical experiment as well as a real-life case study to demonstrate the proposed solution approach. Finally, conclusions are drawn, and future research directions are discussed in Section 5.

## Description of the Chinese freight railway transportation

In this section, we first give a brief introduction to rail freight transportation and then propose the freight train service network design problem. In addition, the tree-shaped path, which is a characteristic pattern in Chinese railway practice, is described.

### Rail freight transportation

Unlike road transportation, in which cargo can be delivered by single trucks, rail freight is generally transported by car groups, i.e., freight trains. Therefore, if a shipment's volume is not large enough to organize an entire direct train from the shipment's origin to its destination, it has to accumulate sufficient goods to be transported to its destination by a direct train. However, in this way, the accumulation process is often time-consuming for shippers and is unfavorable for maintaining steady orders for carriers. To overcome the shortcomings of freight transportation by a direct service, the less-than-train load cargo is generally merged into other shipments and delivered by a sequence of connected train services with several stops in intermediate stations (with coupling and/or uncoupling operations). In Chinese freight railway transportation practice, the less-than-train load cargo is also delivered by connected services. Specifically, these rail cars of cargo are first delivered to their nearby rail classification yards by local train services and blocked with other cars of shipments by humping and marshaling operations. When the total length of these grouped cars reaches the required train length, a new train is formed. Then, the train is dispatched toward its destination station. Fig 1 depicts the situation of the connected services in Chinese freight railway transportation. In the railway

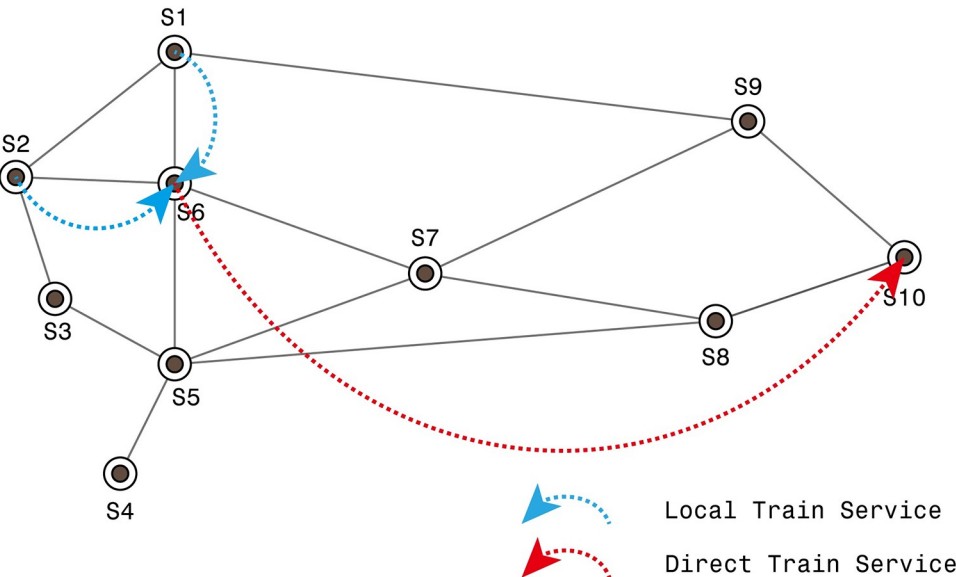

**Fig 1. The connected service strategy of rail freight transportation.**

network consisting of 10 stations/yards, suppose there are three shipments S1→S10, S2→S10 and S6→S10, with volumes of 10 cars per day, 12 cars per day and 28 cars per day, respectively. According to the freight transportation norms, when the volume of the goods reaches at least 50 cars per day, a customized direct train service can be provided daily for these goods. Therefore, these three shipments cannot be transported by a direct service. Using the direct service will result in waiting times of five days, five days, and two days for these three shipments, respectively. In contrast, the connected service strategy is preferable and is more efficient. Specifically, the cars of shipments S1→S10 and S2→S10 can be first transported to S6 and can be grouped with the cars of shipment S6→S10. In this way, the direct train service criterion is satisfied (no less than 50 cars per day), and the direct service from S6 to S10 can be provided each day. As a result, there is no long waiting time for these shipments.

Note that the destinations of the goods loaded in the cars are not necessarily the same as the destination of the train. Once the train arrives at its destination, it is broken up and the cars will get reclassified. If the destination of the cars is different from the uncoupling station, these cars will be grouped again with other cars and be carried to the next station by another new train until reaching their final destination. Clearly, before arriving at the final destination of these cars, they may become consolidated and uncoupled more than once.

## Freight train service network design problem

Consider a railway network consisting of a set of stations and links. Now, we are given a set of shipments that need to be delivered from their origins to their destinations on the railway network. A shipment with a given origin and destination is also called an O-D pair. An O-D pair involves the attributes of origin/destination station, demand volume, goods category, transportation time limit, etc. To transport all the O-D pairs over the railway network, a well-designed freight train service network is needed. The freight train service network plan not only specifies which pair of stations should be provided with direct train services but also determines to which train service(s) each O-D pair should be assigned. Here, a train service has at least the following attributes: origin, destination, train length (measured in cars), route, travel time, traveling speed

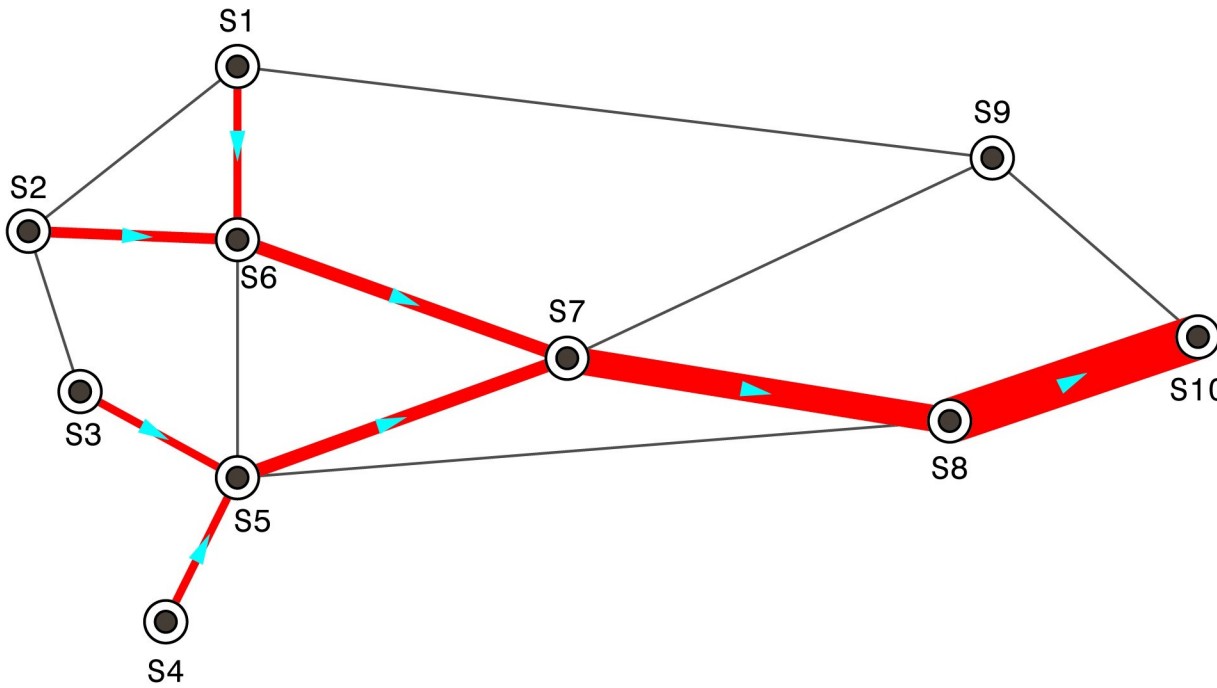

**Fig 2. The tree-shaped path.**

(train class), itinerary length, frequency, and costs. To summarize, the service network design problem (SNDP) for freight railway transportation aims at proposing an optimal service network plan with the objective of minimizing the overall costs while satisfying the operational restrictions (e.g., train length restriction) and predefined constraints (e.g., transportation time limit).

## The tree-shaped path

In China railway operations management, when two or more freight flows to the same destination merge at a railway yard, they will be considered an identical flow and run on the same path during the remainder of the trip, even if they originate in different locations. This consideration forms a unique pattern of flow paths at the macro level, which is defined as the tree-shaped path [32] (or the merging path [33] or common path [34]). Fig 2 shows the tree-shaped path that commonly appears in the context of China's railway freight transportation.

The tree-shaped path phenomenon can be explained from the practical railyard operations for ease of transportation management. The detailed formation mechanism of the tree-shaped path is outside scope of this paper, and we refer interested readers to study [32].

## Optimization model for the freight service network design problem

In this section, we present mathematical formulations for the freight service network design problem. We first provide a detailed explanation of the notations used throughout this paper, followed by an optimization model to mathematically describe the problem.

### Notations

The sets used in this paper are listed in Table 1.

The parameters used in this paper are summarized in Table 2.

The decision variables used in this paper are shown in Table 3.

**Table 1. Sets used in this paper.**

| Symbol | Definition |
|---|---|
| $V$ | Set of stations |
| $C$ | Set of train service classes |
| $P$ | Set of goods categories, the set of goods categories can be divided into two sub-sets: express goods $P^{\text{express}}$ and non-express goods $P^{\text{non}-\text{express}}$, i.e., $P = P^{\text{express}} \cup P^{\text{non}-\text{express}}$ |
| $S$ | Set of train services |
| $D$ | Set of demands, $(o, d, p) \in D$, where $o$, $d$ and $p$ denote the origin station, destination station and goods category of a shipment, respectively |

**Table 2. Parameters used in this paper.**

| Symbol | Definition |
|---|---|
| $\tau_{o,d,p}^{LM}$ | Transportation time limits for demand $(o, d, p)$, where $o$, $d$ and $p$ denote the origin station, destination station and goods category of a shipment, respectively |
| $n_{o,d,p}$ | Volume of demand $(o, d, p)$ in the number of cars, where $o$, $d$ and $p$ denote the origin station, destination station and goods category of a shipment, respectively |
| $l_{i,j,c}$ | Mileage of train service $(i, j, c)$, where $i$, $j$ and $c$ denote the origin station, destination station and class of a service, respectively |
| $\tau_i^{AC}$ | Train accumulation time parameter at station $i$ |
| $\tau_i^{TF}$ | Train transfer time parameter at station $i$ |
| $v_c$ | Average speed of train service class $c$ |
| $m_c$ | Maximal train size (in number of cars) of service class $c$ |
| $c_{i,j,c}^{\text{train}}$ | Train costs of service $(i, j, c)$, where $i$, $j$ and $c$ denote the origin station, destination station and class of a service, respectively |
| $c_p^{\text{car}}$ | Unit car costs of transporting goods category $p$ |

**Table 3. Decision variables used in this paper.**

| Symbol | Definition |
|---|---|
| $x_{i,j,c}^{o,d,p}$ | Binary decision variables, if demand $(o, d, p)$ is assigned to train service $(i, j, c)$, $x_{i,j,c}^{o,d,p} = 1$; otherwise, $x_{i,j,c}^{o,d,p} = 1$, where $o$, $d$ and $p$ denote the origin station, destination station and goods category of a shipment and $i$, $j$ and $c$ denote the origin station, destination station and class of a service, respectively |
| $y_{i,j,c}$ | Integer decision variables, which means the frequency of service $(i, j, c)$, where $i$, $j$ and $c$ denote the origin station, destination station and class of a service, respectively |

## Mathematical formulation

The rail freight service network design problem can be formulated as a mathematical programming model whose objective function and constraints are described as follows.

## Objective function

$$\text{Min } Z = \sum_{(i,j,c)\in S} y_{i,j,c} \cdot c_{i,j,c}^{\text{train}} + \sum_{(o,d,p)\in D,(i,j,c)\in S} x_{i,j,c}^{o,d,p} \cdot l_{i,j,c} \cdot n_{o,d,p} \cdot c_p^{\text{car}} \tag{1}$$

Objective function (1) minimizes overall costs. The overall costs consist of train costs and car costs. The train cost includes the costs of depreciation, maintenance, power, tool and engine drivers [4]. The train cost is associated with train service type (class), train service itinerary length, operation costs of train coupling/uncoupling (i.e., handling costs), etc., while the car

cost includes both car depreciation and maintenance. The car cost is associated with itinerary length, goods volume, goods category, etc.

Subject to

$$\sum_{i \in V, c \in C} x_{i,j,c}^{o,d,p} - \sum_{i \in V, c \in C} x_{j,i,c}^{o,d,p} = \begin{cases} -1 & \text{if } j = o \\ 0 & \text{otherwise} \\ 1 & \text{if } j = d \end{cases} \quad \forall (o,d,p) \in D, j \in V \quad (2)$$

Constraint (2) is the so-called flow balance constraint [35]. It states that all the demand must be satisfied and that no goods are dispersed on the network.

$$\sum_{c \in C} x_{i,j,c}^{o,d,p} \leq 1 \ \forall (o,d,p) \in D, i \in V, j \in V \quad (3)$$

Constraint (3) ensures that each shipment can choose at most a single train service from one station to another station. This constraint states that a single shipment is nonseparable, which is a basic convention in freight transportation of the China railway.

$$\sum_{i \in V, j \in V, c \in C} x_{i,j,c}^{o,d,p} \cdot \frac{l_{i,j,c}}{v_c} + \sum_{i=o, j \in V, c \in C} x_{i,j,c}^{o,d,p} \cdot \tau_i^{AC} \cdot m_c + \sum_{i \neq o, j \in V, c \in C} x_{i,j,c}^{o,d,p} \cdot \tau_i^{TF} \cdot m_c$$
$$\leq \tau_{o,d,p}^{LM} \ \forall (o,d,p) \in D, p \in P^{\text{express}} \quad (4)$$

Constraint (4) ensures that the total transportation time of a shipment should be no longer than the given time limit. This constraint is extremely important for fresh and perishable goods, such as fruits, vegetables and seafood, which are generally transported by express train services. The total transportation time of a shipment consists of three components: the traveling time on railway links, the accumulation time at the origin station and the transfer time at the intermediate station(s). Since the high-speed railway develops rapidly in China during the last decades, a large number of passenger train services shift from normal-speed rail lines to high-speed rail lines. Moreover, express train services are usually dispatched prior to non-express train services as the express train services account for a very low share in the overall railway freight train services. These result in a sufficient network capacity when assigning the express goods to train services and the dispatch of train services are generally not influenced by the network capacity. Consequently, the freight transportation time is heavily reliant on the service classes selected and the number of transfer operations. While in most western railway systems (e.g., the European railway system), most passenger trains and freight trains share the same rail lines [36] and the network capacity becomes an important factor when designing the train service plan. As a result, the freight transportation time depends not only on the assigned services themselves but also on the network capacity implicitly.

$$x_{i,j,c}^{o,d,p} + x_{i,j,b}^{h,k,q} \leq 1 \ \forall (o,d,p), (h,k,q) \in D, i \in V, j \in V, p, q \in P^{\text{non-express}} \quad (5)$$

Constraint (5) is the tree-shaped path constraint. It states that any two shipments (or trains) $(o, d, p)$ and $(h, k, q)$ that are coupled at station $i$ and will be subsequently broken up at station $j$ should take the same train service if the services provided between station $i$ and $j$ use different train paths. As seen from the mathematical formulation, the introduction of the tree-shaped path constraint significantly increases the problem size. The total number of tree-shaped path constraints reaches $|D| \cdot (|D|-1) \cdot |V| \cdot (|V|-1)$. This makes the models harder to solve optimally.

Note that the tree-shaped path constraint holds only for nonexpress goods since express goods usually have predefined tailored train paths and cannot share train services with nonexpress goods both for transportation time limits and cost reasons. The express goods generally

take the express train services running on higher speed train routes, for example, China high-speed railway. In contrast, other goods tend to take the nonexpress train services that operate on normal speed railway networks because these goods do not have strict transportation time limits and the nonexpress train services are much cheaper.

$$\sum_{\forall (o,d,p) \in D} x_{i,j,c}^{o,d,p} \cdot n_{o,d,p} \le m_c \cdot y_{i,j,c} \quad \forall (i,j,c) \in S \tag{6}$$

Constraint (6) links the decision variables $x_{i,j,c}^{o,d,p}$ and $y_{i,j,c}$: the frequency of train service is closely related to the total number of cars using the service.

$$x_{i,j,c}^{o,d,p} \in \{0,1\} \tag{7}$$

$$y_{i,j,c} \in Z \tag{8}$$

Finally, the decision variable domains are specified by Constraints (7) and (8).

## Solution approach

Clearly, the mathematical model proposed in Section 3.2 is integer linear programming (ILP), which is suitable to be directly solved by a standard optimization software package (e.g., an MIP solver such as Gurobi) with its built-in branch-and-cut algorithm framework.

## Numerical experiments

In this section, we conduct a series of numerical experiments to examine the validity of the proposed methodology, including a real-world case study from the China Railway Jinan Group.

## Illustration example

We first adopt an illustration example to demonstrate the overall solution process. The illustrative railway network is depicted in Fig 3. We can see from the figure that the network consists of five stations and five links (link length is marked on the corresponding link).

The station attributes, including station names, train accumulation parameters and train transfer parameters, are shown in Table 4.

The goods categories and the unit car costs of transporting each category of goods are given in Table 5. Without loss of generality, we consider three types of goods, including two types of nonexpress goods (i.e., coal and machinery) and one type of express good (i.e., vegetables).

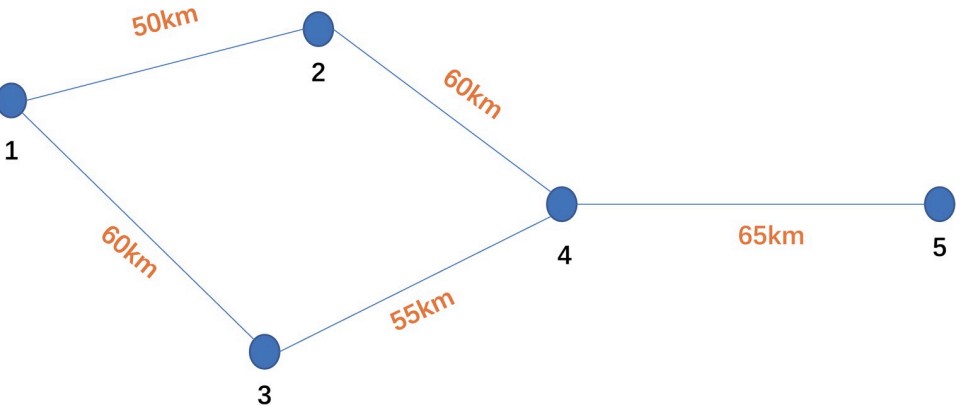

**Fig 3. The illustrative railway network.**

**Table 4. Network stations and accumulation/transfer time parameters.**

| Station | Accumulation time parameter (h) | Transfer time parameter (h) |
|---|---|---|
| 1 | 0.08 | 0.04 |
| 2 | 0.09 | 0.05 |
| 3 | 0.11 | 0.03 |
| 4 | 0.07 | 0.04 |
| 5 | 0.10 | 0.06 |

**Table 5. Goods categories and unit car costs.**

| No. | Category | Express goods? | Unit car costs (CNY[1]/car) |
|---|---|---|---|
| 1 | Coal | No | 0.5 |
| 2 | Machinery | No | 1.5 |
| 3 | Vegetable | Yes | 2.5 |

[1] CNY: Chinese yuan.

**Table 6. Freight demands.**

| No. | Origin | Destination | Category | Volume (cars/day) | Time limit (h) |
|---|---|---|---|---|---|
| 1 | 1 | 2 | Coal | 200 | None |
| 2 | 1 | 4 | Machinery | 30 | 48 |
| 3 | 1 | 5 | Vegetable | 5 | 4 |

**Table 7. Train service classes.**

| Service class | Running speed (km/h) | Train size (cars/train) |
|---|---|---|
| C1 | 160 | 20 |
| C2 | 140 | 30 |
| C3 | 120 | 50 |
| C4 | 100 | 50 |
| C5 | 80 | 50 |

The original freight demands (O-D pairs) are given in Table 6. There are, in total, three shipments that need to be transported, and two of them have transportation time limits.

The freight train service classes are given in Table 7. As seen from the table, there are a total of five service classes considered in the illustration example. These services are characterized by running speed, train size, etc.

In addition, all available freight train services are shown in Table 8. As seen, a total of 50 train services are available. The attributes of service origin/destination stations, service classes, train path length, and unit train service costs for the 50 freight train services are also given in the table.

With the above input data, we solve the mathematical model using the state-of-the-art optimization solver Gurobi 9.1.2 with its default settings. The mathematical model generates a total of 200 decision variables and 541 constraints. The computation is conducted on a laptop with an Intel Core i7-8665U CPU and 16 GB RAM. The model is solved to provide optimum results with a very short solution time of less than one second. The returned optimal objective value is 23000 CNY. The transportation strategy analyzed from the optimal solution is listed in Tables 9 and 10.

**Table 8. Available train services.**

| No. | Origin | Destination | Class | Mileage (km) | Train costs (CNY/train) |
|---|---|---|---|---|---|
| 1 | 1 | 2 | C1 | 50 | 3200 |
| 2 | 1 | 2 | C2 | 50 | 2800 |
| 3 | 1 | 2 | C3 | 50 | 2400 |
| 4 | 1 | 2 | C4 | 50 | 2000 |
| 5 | 1 | 2 | C5 | 50 | 1600 |
| 6 | 1 | 3 | C1 | 60 | 3200 |
| 7 | 1 | 3 | C2 | 60 | 2800 |
| 8 | 1 | 3 | C3 | 60 | 2400 |
| 9 | 1 | 3 | C4 | 60 | 2000 |
| 10 | 1 | 3 | C5 | 60 | 1600 |
| 11 | 1 | 4 | C1 | 115 | 3200 |
| 12 | 1 | 4 | C2 | 115 | 2800 |
| 13 | 1 | 4 | C3 | 115 | 2400 |
| 14 | 1 | 4 | C4 | 110 | 2000 |
| 15 | 1 | 4 | C5 | 110 | 1600 |
| 16 | 1 | 5 | C1 | 180 | 3200 |
| 17 | 1 | 5 | C2 | 180 | 2800 |
| 18 | 1 | 5 | C3 | 180 | 2400 |
| 19 | 1 | 5 | C4 | 175 | 2000 |
| 20 | 1 | 5 | C5 | 175 | 1600 |
| 21 | 2 | 3 | C1 | 115 | 3200 |
| 22 | 2 | 3 | C2 | 115 | 2800 |
| 23 | 2 | 3 | C3 | 115 | 2400 |
| 24 | 2 | 3 | C4 | 110 | 2000 |
| 25 | 2 | 3 | C5 | 110 | 1600 |
| 26 | 2 | 4 | C1 | 60 | 3200 |
| 27 | 2 | 4 | C2 | 60 | 2800 |
| 28 | 2 | 4 | C3 | 60 | 2400 |
| 29 | 2 | 4 | C4 | 60 | 2000 |
| 30 | 2 | 4 | C5 | 60 | 1600 |
| 31 | 2 | 5 | C1 | 115 | 3200 |
| 32 | 2 | 5 | C2 | 115 | 2800 |
| 33 | 2 | 5 | C3 | 115 | 2400 |
| 34 | 2 | 5 | C4 | 115 | 2000 |
| 35 | 2 | 5 | C5 | 115 | 1600 |
| 36 | 3 | 4 | C1 | 55 | 3200 |
| 37 | 3 | 4 | C2 | 55 | 2800 |
| 38 | 3 | 4 | C3 | 55 | 2400 |
| 39 | 3 | 4 | C4 | 55 | 2000 |
| 40 | 3 | 4 | C5 | 55 | 1600 |
| 41 | 3 | 5 | C1 | 120 | 3200 |
| 42 | 3 | 5 | C2 | 120 | 2800 |
| 43 | 3 | 5 | C3 | 120 | 2400 |
| 44 | 3 | 5 | C4 | 120 | 2000 |
| 45 | 3 | 5 | C5 | 120 | 1600 |
| 46 | 4 | 5 | C1 | 65 | 3200 |
| 47 | 4 | 5 | C2 | 65 | 2800 |

(*Continued*)

**Table 8.** (Continued)

| No. | Origin | Destination | Class | Mileage (km) | Train costs (CNY/train) |
|-----|--------|-------------|-------|--------------|-------------------------|
| 48 | 4 | 5 | C3 | 65 | 2400 |
| 49 | 4 | 5 | C4 | 65 | 2000 |
| 50 | 4 | 5 | C5 | 65 | 1600 |

**Table 9. Transportation strategy for each shipment.**

| No. | Shipment | Services used |
|-----|----------|---------------|
| 1 | (1, 2, Coal) | (1, 2, C5) |
| 2 | (1, 4, Machinery) | (1, 4, C5) |
| 3 | (1, 5, Vegetable) | (1, 5, C2) |

**Table 10. Service frequency.**

| No. | Service | Frequency (trains/day) |
|-----|---------|------------------------|
| 1 | (1, 2, C5) | 4 |
| 2 | (1, 4, C5) | 1 |
| 3 | (1, 5, C1) | 1 |

Table 9 shows the detailed transportation strategy for each shipment. As seen from the table, all three involved shipments are transported via direct train services. Among them, shipments (1, 2, Coal) and (1, 4, Machinery) choose lower class train services to save transportation costs, and shipment (1, 5, Vegetable) selects an express train service (140 km/h) due to the demand's transportation time limit. Table 10 summarizes the operating frequencies of the adopted train services. We can see from the table that the frequency of service (1, 2, C5) is four trains per day since the demand volume carried by this service is sufficiently large and the frequencies of both remaining services are only one train per day due to the low demand volume carried by these services.

## Real-world case study

**Data preparation.** Encouraged by the successful computation experience in the illustration of small-scale instances, we are motivated to apply the solution approach to larger-scale real-world instances. The real-world case study data are provided by our collaborator, the China Railway Jinan Group. Officially, abbreviated as CR Jinan or CR-Jinan (formerly, Jinan Railway Administration, which was reorganized as a company in November 2017), the China Railway Jinan Group is a subsidiary company under the jurisdiction of the China Railway (formerly the Ministry of Railway). It supervises the railway network within Shandong Province. It is in charge of the railway with a total length of 8,283.9 kilometers and consists of 295 railway stations.

The real-world case study involves a total of 139 stations that are in charge of CR Jinan. The accumulation/transfer time parameters of the stations can be obtained through the company's statistical operation data.

We collect freight order information from CR Jinan on a typical operation day. The freight order information includes shipment number, shipper, shipment origin station, railway company of the origin station, shipment destination station, railway company of the destination station, goods category, shipment volume, car type needed, transportation time limit and

consignee. On a typical operation day, 562 shipments need to be transported. Among the 562 shipments, there are 96 shipments whose origin station and destination station are both within CR Jinan, and the remainder (466 shipments) have destination stations out of the area of CR Jinan. In this case study, we retain only the shipments where the origin and the destination station are within the area of CR Jinan. To further reduce the problem size and speed up the computation process, we select shipments that have sufficiently large volumes and provide them with direct train services. This approach is also widely adopted by practitioners in the railway industry. Specifically, if the shipment volume reaches more than 50 cars per day, we eliminate it from the shipment dataset and provide the shipment with a direct train service from the shipment's origin to its destination. Consequently, after the data preprocessing procedure, a total of 53 shipments enter the next step of the optimization process.

According to the train service plan designed by CR Jinan, there are 473 available internal train services (where the origin and the destination station are both within the area of CR Jinan) that can be selected by the considered shipments. The available internal train service network is depicted in Fig 4. Due to our confidentiality agreement with CR Jinan, we are not allowed to share the detailed data publicly. To provide basic knowledge of the service data, we only report some of their main features. Among the 473 available train services, there are 25 fast and 448 are ordinary. In the current practice of the China railway system, freight train services are divided into three main categories: express train services with a train running speed of 160 km/h, fast train services with a train running speed of 120 km/h and ordinary services with a train running speed of 80 km/h [37]. Each category of train service has different train sizes, i.e., express, fast and ordinary train services that are usually set as 20, 30 and 50 cars, respectively [38]. Moreover, the longest and shortest train path lengths of the services are 4 km and 805 km, respectively.

In China railway practice, goods are divided into 22 major categories, namely, coal, petroleum, coke, metal ore, steel, nonmetallic ore, mineral building materials, wood, grain, cotton, chemical fertilizer, chemical products, metal, engineering machinery, electrical equipment, fresh products, foods, textile, stationery, medical supplies, and containers, among other goods. Our case study involves most of the goods categories. The transportation time limit restriction

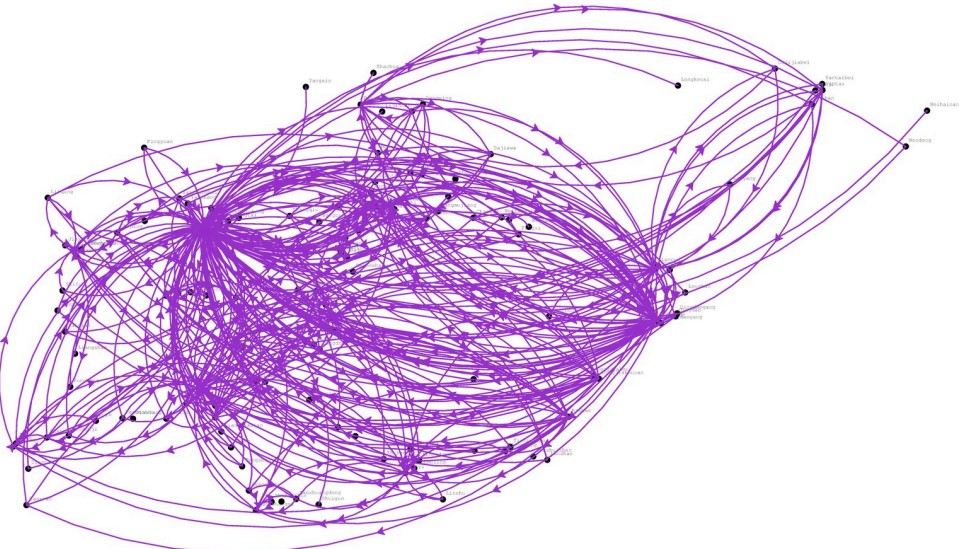

**Fig 4. Available train service network.**

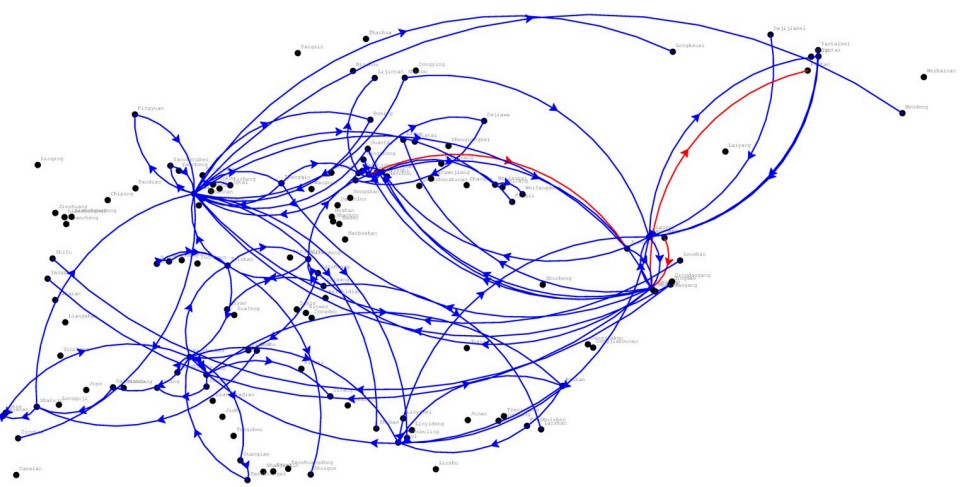

**Fig 5. Optimized train service network.**

generally applies to goods categories of a perishable nature, such as fresh products. However, some shippers also request a strict transportation time limit for their nonperishable goods.

## Computational results

With the above real-life data, we solve the train service network optimization problem using Gurobi on the same laptop mentioned in Section 4.1. Before generating the model, in addition to eliminating the shipments with a sufficiently large volume (see Sec. 4.2.1), we also eliminate the tree-shaped path constraints for the services with the same physical paths to further reduce the model size and accelerate the solution process.

After a computation process of 1,098 seconds, we obtain the optimal solution. The returned optimal objective value (i.e., the overall costs) is 5,900,692.48 CNY. To better visualize the computational results, we depict the optimized train service network in Fig 5. In the figure, the blue line represents the ordinary train service, and the red line represents the fast train service.

A total of 95 services are provided for the considered shipments, including 92 ordinary train services and three fast train services. The frequency of the train services is described as follows. There are 74 train services whose frequency is one train per day, and there are 16 train services with a frequency of two trains per day. In addition, there are four train services that need to dispatch three trains per day, and there is only one train service that transports a sufficiently large number of cars daily with a service frequency of four trains per day.

**Comparisons with manual plans.**  To further highlight the proposed method, we compare the optimal service network with a manual solution developed by the company planners. Table 11 summarizes the comparisons of the solutions by both quality and speed.

The first column of Table 11 represents the solving methods (i.e., the manual method and the proposed method) as well as the changes of solutions by the methods. The second column

**Table 11. Comparisons between the optimal solution and a manual solution.**

| Method | Overall costs (CNY) | Number of train services | Solution time |
|---|---|---|---|
| Manual method | 6467158.43 | 101 | 4–5 hours |
| Proposed method | 5900692.48 | 95 | 1098 seconds |
| Improvements | -566465.95 | -6 | 4–5 hours |
| % Improvements | -8.8% | -5.9% | - |

of the table lists the overall costs using different solving methods. As explained in Section 3.2, the overall costs consist of train costs (including the costs of depreciation, maintenance, power, tool and engine drivers) and car costs (including both car depreciation and maintenance). The third column of the table indicates the total number of provided train services obtain from the solutions. Finally, the fourth column is the solution time required of different methods. From Table 11 we can see that by using the proposed solution approach, the overall costs (objective value) decrease by 566,465.95 CNY (8.8%) and the number of train services is reduced by six (5.9%), while the solution time is reduced by 4–5 hours compared to the manual plan. This indicates that our solution approach outperforms the manual method in terms of both quality and efficiency.

In current practice, the train service network is mainly manually designed by railway dispatchers, which requires several hours of hard effort by a team of highly experienced dispatchers in most Chinese railway companies. Our proposed method is able to provide high-quality solutions within a very short computation time. The results can be used directly by railway dispatchers to save painstaking efforts in developing such plans.

## Conclusions

In this paper, we propose an integer linear programming model for the rail freight service network design problem in the context of the China railway system. The model aims to minimize the overall costs (including the train costs and the car costs) while satisfying a set of practical restrictions and operational requirements. A distinctive constraint considered in the mathematical model is the tree-shaped path, which requires that when two or more freight flows to the same destination merge at a railway yard, they will be considered an identical flow and will run on the same path during the remainder of the trip, even if they originate in different places. The transportation time limit restriction is also considered in the model since our model takes both express goods and nonexpress goods into account.

The effectiveness and efficiency of the optimization model are tested by an artificial instance. Computational results show that the proposed solution approach is able to obtain quality solutions within a very short computation time. Moreover, to further examine the validity of the method, we carry out a real-world case study based on the data provided by our collaborator, CR Jinan. The real-life problem size reaches as many as 139 stations, 473 alternative train services and 562 shipments. The problem is solved to the optimum within 1098 seconds using the state-of-the-art solver Gurobi. Comparisons with manual solutions indicate that our solution approach outperforms the manual method in terms of both quality and computational speed. The computational results can be used directly by railway dispatchers to save painstaking efforts in developing such plans.

As possible future work, the topic of integrating the train routing and timetabling subproblems into the integrated optimization model can be further explored.

## Acknowledgments

The author would like to thank Prof. Xu Wu of Beijing Jiaotong University for her comments and suggestions which improves this paper.

## Author Contributions

**Conceptualization:** Xiaoyu Xin.

**Data curation:** Xiaoyu Xin.

**Formal analysis:** Xiaoyu Xin.

**Investigation:** Xiaoyu Xin.

**Methodology:** Xiaoyu Xin.

**Project administration:** Xiaoyu Xin.

**Resources:** Xiaoyu Xin.

**Software:** Xiaoyu Xin.

**Validation:** Xiaoyu Xin.

**Visualization:** Xiaoyu Xin.

**Writing – original draft:** Xiaoyu Xin.

**Writing – review & editing:** Xiaoyu Xin.

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
