## [Decision Letter · Decision Letter 0]

28 Jul 2022

PONE-D-22-11121Service network design for freight railway transportation: the Chinese casePLOS ONE

Dear Dr. Xin,

Thank you for submitting your manuscript to PLOS ONE. After careful consideration, we feel that it has merit but does not fully meet PLOS ONE’s publication criteria as it currently stands. Therefore, we invite you to submit a revised version of the manuscript that addresses the points raised during the review process.

We look forward to receiving your revised manuscript.

Kind regards,

Dragan Pamucar

Academic Editor

PLOS ONE

Journal Requirements:

“NO authors have competing interests”

3. PLOS requires an ORCID iD for the corresponding author in Editorial Manager on papers submitted after December 6th, 2016. Please ensure that you have an ORCID iD and that it is validated in Editorial Manager. To do this, go to ‘Update my Information’ (in the upper left-hand corner of the main menu), and click on the Fetch/Validate link next to the ORCID field. This will take you to the ORCID site and allow you to create a new iD or authenticate a pre-existing iD in Editorial Manager. Please see the following video for instructions on linking an ORCID iD to your Editorial Manager account: https://www.youtube.com/watch?v=_xcclfuvtxQ.

4. We note that Figure 4 in your submission contain map images which may be copyrighted. All PLOS content is published under the Creative Commons Attribution License (CC BY 4.0), which means that the manuscript, images, and Supporting Information files will be freely available online, and any third party is permitted to access, download, copy, distribute, and use these materials in any way, even commercially, with proper attribution. For these reasons, we cannot publish previously copyrighted maps or satellite images created using proprietary data, such as Google software (Google Maps, Street View, and Earth). For more information, see our copyright guidelines: http://journals.plos.org/plosone/s/licenses-and-copyright.

   a. You may seek permission from the original copyright holder of Figure 4 to publish the content specifically under the CC BY 4.0 license. 

Natural Earth (public domain): http://www.naturalearthdata.com

Reviewers' comments:

Reviewer's Responses to Questions

**Comments to the Author**

1. Is the manuscript technically sound, and do the data support the conclusions?

Reviewer #1: Yes

Reviewer #2: Yes

2. Has the statistical analysis been performed appropriately and rigorously? 

Reviewer #1: Yes

Reviewer #2: N/A

3. Have the authors made all data underlying the findings in their manuscript fully available?

Reviewer #1: Yes

Reviewer #2: Yes

4. Is the manuscript presented in an intelligible fashion and written in standard English?

Reviewer #1: Yes

Reviewer #2: Yes

5. Review Comments to the Author

Reviewer #1: Author present an interesting integer linear programming model for the rail freight service network design problem with the example for China railway system.

In order to be eligible for the publication in Journal the author should more clearly and precisely elaborate on the innovative approach presented in this paper, i.e. how is this model improvement related to the existing and previous models used for this type of problems.

Reviewer #2: In this paper is proposed an integer linear programming model for the rail freight service network design problem with aim to minimize the overall costs in the context of the China railway system.

Тhe author emphasized that problem of rail freight service network design in North American and European freight railway transportation systems are similar in nature with Chines, but the paper only provides a parallel with the American freight transportation system (lines 82-87). Author also mentions on several occasions that problem of rail freight service network design are similar in Europe, but there are no additional explanations of main feature of these problem in Europe (about technology, operation, train routing, wagon management, etc). It would be very useful to add observations and papers related to the problems of design rail freight service and rail network design in the European context. This will support/strengthen the first contribution of this paper (lines 97- 101).

There is lack of some element’s notations. For example, what I, J, C denotes. Please, the author should double-check all the notations (lines 187-193).

When explaining the modeling of the problem, the author introduces limitations in transportation time. However, it cannot be determined whether these are time restrictions due to the applied technology or whether they are restrictions due to network capacity (lines 213-217).

CONCLUSION:

This manuscript is technically sound. It is in an intelligible fashion and written in standard English (which is approved by certificate as well). Finally, the manuscript is well organized and written clearly enough to be accessible to non-specialists and itself show significant potential and the author is encouraged to resubmit a revised version.

6. PLOS authors have the option to publish the peer review history of their article (what does this mean?). If published, this will include your full peer review and any attached files.

Reviewer #1: No

Reviewer #2: No

---

## [Author Response · Author response to Decision Letter 0]

22 Aug 2022

Please see the attached "Response to Reviewers" file for our detailed response and revisions with respect to the reviewers' comments.

---

## [Decision Letter · Decision Letter 1]

4 Sep 2022

PONE-D-22-11121R1Service network design for freight railway transportation: the Chinese casePLOS ONE

Dear Dr. Xin,

Thank you for submitting your manuscript to PLOS ONE. After careful consideration, we feel that it has merit but does not fully meet PLOS ONE’s publication criteria as it currently stands. Therefore, we invite you to submit a revised version of the manuscript that addresses the points raised during the review process.

We look forward to receiving your revised manuscript.

Kind regards,

Dragan Pamucar

Academic Editor

PLOS ONE

Journal Requirements:

Reviewers' comments:

Reviewer's Responses to Questions

**Comments to the Author**

1. If the authors have adequately addressed your comments raised in a previous round of review and you feel that this manuscript is now acceptable for publication, you may indicate that here to bypass the “Comments to the Author” section, enter your conflict of interest statement in the “Confidential to Editor” section, and submit your "Accept" recommendation.

Reviewer #1: All comments have been addressed

Reviewer #2: All comments have been addressed

2. Is the manuscript technically sound, and do the data support the conclusions?

Reviewer #1: Yes

Reviewer #2: Yes

3. Has the statistical analysis been performed appropriately and rigorously? 

Reviewer #1: N/A

Reviewer #2: Yes

4. Have the authors made all data underlying the findings in their manuscript fully available?

Reviewer #1: Yes

Reviewer #2: Yes

5. Is the manuscript presented in an intelligible fashion and written in standard English?

Reviewer #1: Yes

Reviewer #2: Yes

6. Review Comments to the Author

Reviewer #1: Author have addressed the comments and provided the response, claiming that the organizational difference between China and EU or USA makes this problem solving model with unique and innovative approach.

I accept the response.

Reviewer #2: Thank you very much for the explanation you gave, but you did not improve the paper considering those responses to the comments.

I suggest that you include a detailed explanation about transport time in the paper, because it precisely indicates the difference between the European and Chinese freight systems. Also expand the explanation of the difference between the European and Chinese systems in the text.

Additionaly, for comparisons between the optimal solution and a manual solution are used results of overall costs and solution time (lines 356-356). For solution time, operational times for defining train routes by highly experienced dispatchers (skill, competence) and computer speed (depends on the type of computer, software solution and amount of data) are compared. It is not clear what is included in overall cost figure. Are these only transportation costs or they are also other costs? Tariffs are the same because the same type and quantity of goods are transported. The author should explain in the text and strengthen results in this sense.

7. PLOS authors have the option to publish the peer review history of their article (what does this mean?). If published, this will include your full peer review and any attached files.

Reviewer #1: No

Reviewer #2: No

---

## [Author Response · Author response to Decision Letter 1]

21 Sep 2022

Please see the attached "Response to Reviewers" file for our detailed response and revisions with respect to the reviewers' comments.

---

## [Decision Letter · Decision Letter 2]

12 Oct 2022

Service network design for freight railway transportation: the Chinese case

PONE-D-22-11121R2

Dear Dr. Xin,

We’re pleased to inform you that your manuscript has been judged scientifically suitable for publication and will be formally accepted for publication once it meets all outstanding technical requirements.

Kind regards,

Dragan Pamucar

Academic Editor

PLOS ONE

Additional Editor Comments (optional):

Reviewers' comments:

Reviewer's Responses to Questions

**Comments to the Author**

1. If the authors have adequately addressed your comments raised in a previous round of review and you feel that this manuscript is now acceptable for publication, you may indicate that here to bypass the “Comments to the Author” section, enter your conflict of interest statement in the “Confidential to Editor” section, and submit your "Accept" recommendation.

Reviewer #2: All comments have been addressed

2. Is the manuscript technically sound, and do the data support the conclusions?

Reviewer #2: Yes

3. Has the statistical analysis been performed appropriately and rigorously? 

Reviewer #2: N/A

4. Have the authors made all data underlying the findings in their manuscript fully available?

Reviewer #2: Yes

5. Is the manuscript presented in an intelligible fashion and written in standard English?

Reviewer #2: Yes

6. Review Comments to the Author

Reviewer #2: (No Response)

7. PLOS authors have the option to publish the peer review history of their article (what does this mean?). If published, this will include your full peer review and any attached files.

Reviewer #2: No

---

## [Editor Report · Acceptance letter]

18 Nov 2022

PONE-D-22-11121R2 

Service network design for freight railway transportation: the Chinese case 

Dear Dr. Xin:

I'm pleased to inform you that your manuscript has been deemed suitable for publication in PLOS ONE. Congratulations! Your manuscript is now with our production department. 

Kind regards, 

on behalf of

Dr. Dragan Pamucar 

Academic Editor

PLOS ONE